# Risk Factors for Acute Postoperative Delirium in Cardiac Surgery Patients >65 Years Old

**DOI:** 10.3390/jpm12091529

**Published:** 2022-09-18

**Authors:** Eleni Spiropoulou, George Samanidis, Meletios Kanakis, Ioannis Nenekidis

**Affiliations:** 1Department of Cardiac Surgery Intensive Care, Onassis Cardiac Surgery Center, 17674 Athens, Greece; 2Department of Adult Cardiac Surgery, Onassis Cardiac Surgery Center, 17674 Athens, Greece; 3Onassis Cardiac Surgery Center, 356 Leoforos Syggrou, 17674 Athens, Greece; 4Department of Pediatric and Congenital Heart Surgery, Onassis Cardiac Surgery Center, 17674 Athens, Greece

**Keywords:** cardiac surgery, acute delirium, hypoxemia, alcohol use, intensive care unit

## Abstract

**Background:** Acute postoperative delirium is the most common neuropsychiatric disorder in cardiac surgery patients in the intensive care unit (ICU). The purpose of this study was to evaluate the possible risk factors of postoperative delirium (POD) for cardiac surgery patients in the ICU. **Materials and Methods:** The study population was composed of 86 cardiac surgery patients managed postoperatively in the cardiac surgery ICU. Presence of POD in patients was evaluated by the CAM-ICU scale. **Results:** According to the CAM-ICU scale, 22 (25.6%) patients presented POD; history of smoking, alcohol use, COPD, and preoperative permanent atrial fibrillation were associated with POD (for all, *p* < 0.05). The type of cardiac surgery operations, type of analgesia, and red blood cell transfusion in the ICU were not associated with POD (*p* > 0.05), while cardiac arrhythmia in the ICU, hypoxemia in the ICU after extubation (pO_2_ < 60 mmHg), and heart rate after extubation were predisposing factors for POD (for all, *p* < 0.05). Multivariable logistic regression analysis (adjusted to risk factors) showed that hypoxemia after extubation (OR = 20.6; 95%CI: 2.82–150), heart rate after extubation (OR = 0.95; 95% CI: 0.92–0.98), and alcohol use (OR = 74.3; 95%CI: 6.41–861) were predictive factors for acute postoperative delirium (for all, *p* < 0.05). **Conclusion:** Alcohol use and respiratory dysfunction before and after heart operation were associated with acute postoperative delirium in cardiac surgery ICU patients.

## 1. Introduction

Cognitive disorders after cardiac surgery can be presented within a spectrum of different variations in mental and emotional pathology, ranging from a mild event of temporary loss of concentration to multiple episodes of acute delirium, a condition which can jeopardize the course of a smooth postoperative outcome [1].

In fact, the term postoperative delirium (POD) is defined as temporary in most cases, and as a fluctuated disorder of consciousness presenting early in the post-surgery aftercare period. It usually appears in elderly people and in most severe heart disease patients with heavy concomitant pathology [2]. There is also a decrease in the ability to think, perceive, and recall memory, as well as intense psychomotor stimulation. The keystone symptom of POD is the disturbance of consciousness highlighted with repeated advanced alert activity just before each time the patient reaches a blurry and foggy communication environment. The classification of this neuropsychiatric condition is based on different modalities of alertness in close relation to the level of consciousness described and this basically is characterized as hyperkinetic, hypokinetic, or mixed [3].

Thus, hyperkinetic delirium is a condition of stressful behavior attached to the feeling of agony and anxiety and, in many cases, is characterized by additional elements of hallucinations and illusions. In addition, the patient may experience hallucinations (visual and auditory) and the stimulation may be due to them. The patient is usually scared and potentially aggressive. On the other side, the hypokinetic form of delirium is presented with signs of apathy, withdrawal, and leveling of emotions. The patient is usually lethargic with somnolent behavior and reduced response to stimuli. This condition resembles depression; therefore, the differential diagnosis depends on specific details derived from the psychokinetic deceleration status of the patient. Finally, the mixed type is a variety of fluctuations and transitions in between a hyper- and hypokinetic state.

The top marks of predisposing factors for developing POD include increased age, decreased cognitive function, alcohol and drug abuse, male sex, systemic adverse effects of irreversible end stage diseases, genetic predisposing, electrolyte disturbances and, potentially, depression and dementia [4,5,6,7,8]. The onset of POD is associated with an increased length of stay in-hospital and the ICU, postoperative rehabilitation, and mortality after discharge from the hospital, as well as the subsequent functional impoverishment of patients regardless of the presence or absence of dementia.

The purpose of this study was to evaluate the possible risk factors for POD after cardiac surgery in patients who were postoperatively treated in the ICU. 

## 2. Patients and Methods

### 2.1. Study Population

The study included 86 patients who underwent cardiac surgery operations (coronary bypass grafting (CABG), heart valve surgery, or combined operations) via cardiopulmonary bypass and who were postoperatively treated in the cardiac surgery ICU. Inclusion criteria for the study were: (1) extubated patients, (2) patients with no known psychiatric disease, (3) patients could speak and understand the language, and (4) the age of patients was > 65 years old. The informed consent was obtained from the patients included in this study. This study has been approved by the Hospital Institutional Review Board (532/01-10-2014).

Present or absent POD was estimated using the confusion assessment methods-intensive care unit (CAM-ICU) scale [9]. The level of arousal and repression was measured using the Richmond Agitation-Sedation Scale (RASS), which is a 10-point scale ranging from +4 to −5, with a RASS score of 0 indicating a calm and awake patient. The diagnosis of POD was established according to the Diagnostic and Statistical Manual of Mental Disorders (DSM-IV) and if the patients met the following criteria based on the CAM-ICU scale: (1) the mental status of patients was changed acutely, accompanied with (2) inattention; (3) disorganization of thinking; or (4) an altered level of consciousness. 

### 2.2. Statistical Analysis

Continuous variables were presented by mean ± standard deviation (SD), while nominal variables were presented by number (n) and percentage (%). The normality of the distribution of variables was evaluated using the Kolmogorov–Smirnov test and Q-Q plot. Parametric (Student’s *t*-test) or non-parametric tests (Mann–Whitney test, Kruskal–Wallis test, Chi-squared test, and Fisher’s exact test) were implemented for the data analysis, depending on the normality of variable distribution. Spearman (r_s_) or Pearson (r) correlation tests were implemented for data analysis. Binary univariable and multivariable logistic regression analysis was implemented to identify the risk factors for POD and the effect size of risk factors to POD was expressed by odds ratio (OR). For logistic regression analysis, the Hosmer–Lemeshow test was used for goodness of fit. The confidence interval was set at 95% (95% confidence interval). The statistically significant difference was considered *p* < 0.05. IBM SPSS Statistics for Windows, version 25 (IBM Corp., Armonk, NY, USA) was used for data analysis. 

## 3. Results

### 3.1. Demographic Characteristics

The study population consisted of 86 patients in which 29 patients (33.7%) were women. Regarding education, 41.9% of patients were primary school graduates while 11.6% of patients were higher education. A total of 16.3% of patients lived alone. The most common comorbidities were hypertension in 87.2% of patients and DM in 38.4% of patients. Forty-four patients had undergone previous surgery (not cardiac surgery), and 64% of patients report that they do sedentary work. Smoking and alcohol use was reported in 16.3% and 8.1% of patients, respectively. Other demographics and preoperative characteristics of patients are presented in Table 1 and Table 2.

### 3.2. Perioperative Details

Regarding the type of surgery, 41.9% of patients had undergone heart valve surgery, while the most common duration of operation (3–6 h) was recorded in 51 (59.3%) patients, and the duration of CPB lasting more than 3 h was observed in 11 (12.8) patients. Regarding intraoperative blood gases, 16.3% of the patients had a pH < 7.35 and 55.9% of patients had a pH = 7.36–7.45, while in 27.8% of patients, the pH range was 7.46–7.53. As for reference pCO_2_, 57% of patients had < 34 mmHg, 39.5% had 35 mmHg–45 mmHg, and 3.5% had 46 mmHg–48 mmHg. In terms of the type of analgesic administered, 64 (74.4%) patients used paracetamol + morphine, while paracetamol + morphine + midazolam were administered for 9 (10.5%) patients. According to the CAM-ICU scale, 22 (25.6%) patients postoperatively presented acute postoperative delirium in ICU. Additional perioperative details are listed in Table 2.

### 3.3. Comparison of Two Groups with and without Acute Postoperative Delirium

Analysis of our data showed that gender, patient age being <74 and ≥75 years old, education level, hypertension, diabetes mellitus, and patient living alone were not associated with POD (for all, *p* > 0.05) (Table 3). On the other hand, history of smoking, alcohol use, COPD, and preoperative permanent atrial fibrillation were associated with POD (for all, *p* < 0.05) (Table 3). In addition, the type of cardiac surgery operations, type of analgesia, and red blood cell transfusion were not associated with acute delirium (*p* > 0.05), while cardiac arrhythmia in the ICU, postoperative hypoxemia in the ICU after extubation (pO_2_ < 60 mmHg), and heart rate were predisposing factors for acute delirium (for all, *p* < 0.05). Other intraoperative and postoperative results are shown in Table 4. A positive CAM-ICU value was associated with heart rate (*p* = 0.04), while the age of patients, intubation time in the ICU, response time, and operation time were not associated with CAM-ICU (*p* > 0.05).

### 3.4. Univariable and Multivariable Logistic Regression Analysis

Univariable logistic regression analysis showed that permanent atrial fibrillation (*p* = 0.04, OR = 4.4; 95% CI: 1.06–18.2), smoking (*p* = 0.02, OR = 3.8; 95% CI: 1.15–12.5), alcohol use (*p* = 0.005, OR = 23.6; 95% CI: 2.65–210.4), intubation time in ICU (*p* = 0.02, OR = 1.04; 95% CI: 1.0–1.07), cardiac arrhythmia in the ICU (*p* = 0.03, OR = 3.3; 95% CI: 1.15–9.22), and hypoxemia in the ICU after extubation (pCO_2_ < 60 mmHg) (*p* = 0.008, OR = 7.6; 95% CI: 1.71–33.8) were predictive factors for postoperative delirium. Multivariable regression logistic analysis evaluated the risk factors for POD. Multivariable logistic regression analysis (adjusted to risk factors) showed that hypoxemia after extubation (OR = 20.6; 95%CI: 2.82–150), heart rate after extubation (OR = 0.95; 95% CI: 0.92–0.98), and alcohol use (OR = 74.3; 95%CI: 6.41–861) were predictive factors for acute postoperative delirium (*p* < 0.05 for all).

## 4. Discussion

The onset of acute postoperative delirium (POD) in ICU patients concerns the health professionals who are caring for these patients. Spear reported an increased risk of mortality; institutionalization; and dementia; independent of age; and comorbidities in patients who had versus who did not have delirium, and the mean duration of observation was 22.7 months [10]. In patients with existing disorders of cognitive functions, the rate of delirium developed with high risk and accounted for approximately in 22–89% of patients [11]. A range of 15–35% of patients presenting with POD belonged to the elderly group [12]. In the intensive care units, much higher rates of delirium have been observed, ranging from 70% to 87% of patients [6]. In addition, in patients with malignant diseases, the chances of developing delirium during the course of the disease are 11–35%, while in the final stages of the disease (as in the last weeks of life), in severe clinical conditions, its occurrence is the most common complication, which reaches 85–88% [13]. The same high percentage of 88% is observed in palliative care units of the general type [14]. In the present study, with a sample of 86 cardiac surgery patients, an attempt was made to investigate, analyze, and detect the predisposing factors that contribute to the occurrence of acute POD. According to our results, 25.6% of patients developed acute POD. On the other hand, many studies have reported that POD affects the long-term survival of patients who have undergone cardiac surgery. Gottesman et al. found that the survival rate (over 10 years) of cardiac surgery patients without POD was higher than the patients with delirium [15].

Different risk factors present to predict and identify POD and among these are alcohol use, chronic obstructive disease, smoking, number of drugs used, electrolyte disturbance, and type of anesthesia used [4,8,16]. 

Regarding alcohol use, it seems to be positively correlated with POD surgery patients. The results of one study showed higher rates of POD in the user group [8]. Chronic obstructive pulmonary disease is a respiratory disease with its main feature being the obstruction of the airways of the respiratory system resulting in shortness of breath and hypoxemia. Hypoxemia can lead to decreased acetylcholine levels, making individuals susceptible to delirium. In the present study, it appears to be a predisposing risk factor for acute postoperative delirium. Similarly, Szylińska et al. investigated predisposing factors for delirium in cardiac surgery patients, namely preoperative COPD. They revealed that 22.97% of patients with COPD were diagnosed with POD and they concluded that COPD is an important predisposing risk factor for the development of POD in cardiac surgery patients [17]. In our study, for the patients with preoperative COPD, postoperative delirium was diagnosed in 27.3% of patients, with 9.4% of patients not experiencing delirium (*p* < 0.05).

Postoperative hyponatremia appears to be positively associated with the occurrence of acute POD. This conclusion agrees with Smutler et al. in a prospective study of 142 patients (≥70 years) who underwent cardiac surgery [18]. They found that sodium concentration was associated with delirium. The same results were presented by Onuma et al. in patients who underwent spinal surgery at ages > 75 years old [19]. Regarding the type and number of anesthetics administered in the present study, it does not appear to be a predisposing factor for the occurrence of acute postoperative delirium. In addition, no difference in POD was observed in a study by Shin et al. that enrolled 534 cardiac surgery patients [20]. The study showed that sevoflurane with dexmedetomidine and propofol did not affect the development of POD. A recent randomized controlled trial by Momeni et al. presented no benefits of combining propofol plus dexmedetomidine versus propofol alone to prevent POD in cardiac surgery patients [21]. On the other hand, Subramanian et al. presented in their study that postoperative analgesia by postoperative intravenous acetaminophen administration combined with propofol or dexmedetomidine reduced in-hospital delirium [1]. Reports have been made of the effect of nicotine on chronic smokers and its association with delirium. According to the results of the study, chronic smokers are more likely to develop postoperative delirium due to nicotine dependence and abrupt cessation, which is consistent with the results of a study by Galyfos et al. [22]. Miyazaki et al. identified that smoking is a predisposing factor for POD in patients who underwent off-pump CABG [23]. In our study, smoking was associated with POD (*p* = 0.03).

The results from the postoperative evaluation of patients in terms of heart rate showed that arrhythmia was positively associated with the occurrence of acute postoperative delirium [24]. A consequence of arrhythmia, particularly the bradycardia, is a low cardiac output syndrome and cerebral hypoperfusion with hypoxemia. There were different types of bradycardias encountered postoperatively. These were usually sinus bradycardia, atrial fibrillation with low frequency, nodal rhythm, and second and third degree atrioventricular block. The causes of bradycardia can be varied, such as heart attack or ischemia, heart valve replacement, ASD correction, electrolyte disturbances, or even pharmacological etiology. Sometimes the consequence of bradycardia was hemodynamic instability with severe hypotension (SAP < 80 mmHg). In conclusion, bradycardia presenting in patients as a postoperative complication with consequent low cardiac output was a predisposing factor for the occurrence of POD [25].

## 5. Study Limitations

The small number of patients included in this study may affect our results and possible risk factors may have been underestimated. In addition, it is a single-center study and focuses on ICU patients. All patients included in this study were only cardiac surgery patients and extracted results should not be adapted for ICU patients with other pathologies in the general population. All patients underwent cardiac surgery via cardiopulmonary bypass and alcohol abuse was not recorded in the study population. 

## 6. Conclusions

After analyzing our data, we concluded that preoperative COPD, smoking, and alcohol use were predisposing factors for the appearance of POD in ICU patients who underwent cardiac surgery operations. 

## Figures and Tables

**Table 1 jpm-12-01529-t001:** Demographic characteristics of patients. Number = N or n.

Variable	Total Number of Patients N = 86 (%)
**Gender, Woman, n (%)**	29 (33.7)
**Age, years old**	
<74 ≥75	48 (55.8) 38 (54.2)
**Body mass index, n (%)**	
Normal Overweight Obesity	17 (19.7) 50 (58.1) 19 (22.2)
**Patient lives alone, n (%)**	14 (16.3)
**Education**	
Primary Secondary Higher	36 (41.9) 40 (46.6) 10 (11.6)
**Profession, n (%)**	
Manual work Sedentary work	31 (36) 55 (64)
**Smoking**	14 (16.3)
**Mobility**	
Good Poor	84 (97.7) 2 (2.3)
**Walking, n (%)**	49 (57)
**Alcohol use, n (%)**	7 (8.1)
**Drug use, n (%)**	1 (1.2)

**Table 2 jpm-12-01529-t002:** **Preoperative and intraoperative details.** Number = N or n; chronic obstructive pulmonary disease = COPD; coronary artery bypass grafting = CABG; intensive care unit = ICU; confusion assessment methods-intensive care unit = CAM-ICU.

Variable	Total Number of Patients N = 86 (%)
**Hypertension, n (%)**	75 (87.2)
**Diabetes mellitus, n (%)**	33 (38.4)
**COPD, n (%)**	12 (14)
**Hyperlipidemia, n (%)**	31 (36)
**Chronic renal insufficiency without hemodialysis, n (%)**	9 (10.5)
**Preoperative cardiac arrhythmia, n (%)**	9 (10.5)
**Thyroid disease, n(%)**	16 (18.6)
**Previous operation (other than cardiac surgery), n (%)**	44 (51.2)
**Depression**	6 (7)
**Type of operation, n (%)**	
Heart valve CABG Combined	36 (41.9) 24(27.9) 26 (30.2)
**Time of operation, n (%)**	
<3 h 3–6 h >6 h	32 (37.2) 51 (59.3) 3 (3.5)
**Cardiopulmonary bypass time, n (%)**	
<3 h >3 h	75 (87.2) 11 (12.8)
**Intubation time, n (%)**	
<24 h 24–48 h >48 h	77 (89.5) 5 (6.8) 4 (4.7)
**Blood transfusion in ICU, n (%)**	15 (17.4)
**Cardiac arrhythmia in ICU, n (%)**	23 (26.7)
**Hypoxemia after extubation in ICU (pO_2_ < 60 mmHg), n (%)**	9 (10.5)
**Acute postoperative delirium (CAM-ICU), n (%)**	22 (25.6)

**Table 3 jpm-12-01529-t003:** **Comparison of demographic characteristics between two groups with and without acute postoperative delirium.** Chronic obstructive pulmonary disease = COPD; number = N or n; standard deviation = SD. * Statistical significance *p* < 0.05.

Variables	Acute Postoperative Delirium		*p*-Value
	Yes N = 22 Patients (%)	No N = 64 Patients (%)	
**Age, years old ± SD**	74.6 ± 5.7	72.4 ± 5.3	0.11
**Body mass index, kg/m^2^ ± SD**	27.8 ± 4.2	28.2 ± 3.5	0.69
**Gender, n (%)**			
Man Woman	14 (63.6) 8 (36.4)	43 (67.2) 21 (32.8)	0.47
**Age, years old, (%)**			
<74 ≥75	9 (40.9) 13 (59.1)	39 (60.9) 25 (39.1)	0.08
**Body mass index, n (%)**			
Normal Overweight Obesity	6 (27.3) 12 (54.5) 4 (18.2)	11 (17.2) 38 (59.4) 15 (23.4)	0.57
**Patient lives alone, n (%)**	3 (13.6)	11 (17.2)	0.49
**Education, n (%)**			
Primary-Secondary Higher	11(50) 11 (50)	29 (45.3) 35 (54.7)	0.44
**Profession, n (%)**			
Manual work Sedentary work	8 (36.4) 14 (63.6)	23 (35.9) 41 (64.1)	0.58
**Smoking, n (%)**	7 (31.8)	7 (10.9)	**0.02 ***
**Alcohol use, n (%)**	6 (27.3)	1 (1.6)	**0.001 ***
**Hypertension, n (%)**	19 (86.4)	56 (87.5)	1.00
**Diabetes mellitus, n (%)**	8 (36.4)	25 (39.1)	0.82
**COPD, n (%)**	6 (27.3)	6 (9.4)	**0.03 ***
**History of malignancy, n (%)**	4 (18.2)	3 (4.7)	0.05
**Hyperlipidemia, n (%)**	9 (40.9)	22 (34.4)	0.38
**Permanent atrial fibrillation, n (%)**	5 (22.7)	4(6.3)	**0.02 ***
**Previous surgery (non-cardiac), n (%)**	8 (36.4)	36 (56.3)	0.10
**Carotid disease, n (%)**	2 (9.1)	2 (3.1)	0.25
**Chronic renal failure, n (%)**	3 (13.6)	6 (9.4)	0.57

**Table 4 jpm-12-01529-t004:** **Comparison of perioperative details between two groups with and without acute postoperative delirium.** Number = N or n; standard deviation = SD. * Statistical significance *p* < 0.05.

Variables	Acute Postoperative Delirium		*p*-Value
	Yes N = 22 Patients	No N = 64 Patients	
**Type of operation, n (%)**			
Heart valve CABG Combined	9 (40.9) 5 (22.7) 8 (36.4)	27 (42.2) 21 (32.8) 16 (25.0)	0.51
**Operation time, hours ± SD**	3.8 ± 1.3	3.3 ± 1	0.12
**Cardiopulmonary bypass time, hours ± SD**	2.0 ± 1.0	1.7 ± 0.9	0.24
**pH (intraoperative) ± SD**	7.4 ± 0.062	7.4 ± 0.056	0.92
**pCO_2_ (intraoperative), mmHg ± SD**	34.5 ± 5.5	34 ± 4.4	0.70
**pO_2_ (intraoperative), mmHg ± SD**	223.5 ± 81.1	232.1 ± 103.2	0.85
**Level of K^+^(intraoperative), mmol/L ± SD**	4.6 ± 0.59	4.5 ± 0.59	0.25
**Level of Na^+^ (intraoperative), mmol/L ± SD**	135.1 ± 3.7	137 ± 4.18	**0.03 ***
**Level of Ca^++^ (intraoperative), mmol/L ± SD**	1.1 ± 0.54	1 ± 0.18	0.56
**Level of Mg^++^ (intraoperative) ± SD**	2.2 ± 0.62	2.3 ± 0.56	0.86
**Level of glucose (intraoperative), mg/dL ± SD**	147.7 ± 54.37	145.7 ± 38.04	0.61
**Level of lactate (intraoperative), mmol/L ± SD**	1.8 ± 0.76	2.2 ± 1.01	0.05
**Intubation time in ICU, hours ± SD**	23.8 ± 22.95	14.3 ± 9.21	0.18
**Response time, min ± SD**	3.7 ± 6.68	1.4 ± 0.73	0.07
**pH (2 h after extubation) ± SD**	7.4 ± 0.05	7.4 ± 0.035	0.54
**pCO_2_ (2 h after extubation), mmHg ± SD**	39.5 ± 4.48	38.1 ± 3.44	0.24
**pO_2_ (2 h after extubation), mmHg ± SD**	108.6 ± 28.1	113 ± 34.42	0.92
**Level of K^+^ (2 h after extubation), mmol/L ± SD**	4.4 ± 0.37	4.4 ± 0.35	0.98
**Level of Na^+^ (2 h after extubation), mmol/L ±SD**	140 ± 2.69	137.7 ± 13.2	0.55
**Level of Ca^++^ (2 h after extubation), mmol/L ± SD**	1 ± 0.2	1.1 ± 0.24	0.75
**Level of Mg^++^ (2 h after extubation) ± SD**	2.2 ± 0.61	2.2 ± 0.66	0.88
**Level of glucose (2 h after extubation), mg/dL ± SD**	176.2 ± 45.65	173.2 ± 31.24	0.87
**Level of lactate (2 h after extubation), mmol/L ± SD**	1.51 ± 0.74	1.7 ± 0.87	0.36
**Systolic blood pressure (4 h after extubation), mmHg ± SD**	132 ± 12.77	127.6 ± 12.93	0.12
**Diastolic blood pressure(4 h after extubation), mmHg ± SD**	61.8 ± 7.64	62.3 ± 8.75	0.93
**Heart rate (4 h after extubation) ± SD**	77.1 ± 26.13	86.8 ± 15.68	**0.04 ***
**Temperature (4 h after extubation), °C ± SD**	37.2 ± 0.41	42.5 ± 42.48	0.84
**Hypoxemia after extubation (pO_2_ < 60 mmHg),** **n (%) ± SD**	6 (27.3)	3 (4.7)	**0.008 ***
**Respiratory rates per minute (4 h after extubation) ± SD**	19.5 ± 3.51	18.63 ± 5.02	0.40
**Red blood cells transfusion in ICU, n (%)**	2 (9.1)	13 (20.3)	0.23
**Cardiac arrhythmia in ICU, n (%)**	10 (45.5)	13 (20.3)	**0.02 ***

## Data Availability

Data available from the authors upon request.

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
