# Peer review of "Risk Factors for Acute Postoperative Delirium in Cardiac Surgery Patients >65 Years Old"

_jpm, 2022, doi:10.3390/jpm12091529_

Round 1

Reviewer 1 Report

Dear Writers and editor,

The article proposed  by Samanidis is very interesting and well built, moreover there are few articles well processed  like this manuscript.

I have just 2 doubt: are you sure to compare open surgery like valve surgery with closed surgey like grafting?

should you consider snari and ssri are unprotectoive or protrctive?

Author Response

Dear Editors and Reviewers,

Thank you for your interesting comments to our manuscript. We try to answer to reviewers comments and we hope that our manuscript will be suitable for publication in your prestigious Journal. 

Although the cardiac operations are used worldwide for cardiac pathologies correction, the exact causes of postoperative neurological and psychiatric dysfunctions are not clear. Hypothermia, hypoperfusion and intermittent circulatory arrest contribute to temporary or permanent neurological dysfunctions. Few studies with analysis of risk factors for postoperative acute delirium were reported in the literature. In our study, we presented the risk factors for acute postoperative delirium in cardiac surgery patients.

Thank you.

Sincerely,

George Samanidis, MD, PhD
Cardiac Surgeon

Reviewer 1

Comments and Suggestions for Authors

Dear Writers and editor,

Comment 1 The article proposed by Samanidis is very interesting and well built, moreover there are few articles well processed like this manuscript.
Answer 1 Thank you

I have just 2 doubt: 
Comment 2: are you sure to compare open surgery like valve surgery with closed surgery like grafting?
Answer 2: Thank you for your comment. We did not include patients who underwent cardiac surgery without cardiopulmonary bypass. In our study included the pts who underwent cardiac operations by cardiopulmonary bypass only. 

Comment 3: should you consider snari and ssri are un-protective or protective?
Answer 3: Thank you for your comment. The patients who had any preoperative psychiatric and neurological dysfunctions or received any neurtropic drugs did not include in our study. 

Reviewer 2

Comments and Suggestions for Authors

Comment 1: As a general observation, the title and content of the paper does not fit very well in the Special Issue theme.
Answer 1: Thank you for your comment.  Our manuscript meets the criterias of the special issue theme. 
1) The title and content of the manuscript regard specific postoperative complication after advance cardiac pathology treatment. 
2) Cardiac surgery consist par excellence advance cardiac disease. 
3) We present a rare postoperative complication after cardiac surgery which needs specific treatment and management in specific group of pts 
4) In our study, we included the patients who was >65 y.o. It is specific group of patients who underwent cardiac surgery. 
5) Although postoperative delirium in cardiac surgery pts is observed rarely, on next years the number of older pts who underwent cardiac surgery will be increased and the knowledge of exact risk factors and mechanism to appearance the post-op delirium may help to manage and treat these patients peri-op. 
I think that our manuscript meets criteria of special issue
Change 1: We change the title.

Comment 2: The study you have presented was mend to enlighten other risk factors for POD, than those already presented in the literature from years 2000-2010. I expected from the team of authors to bring something new, or at least presented from another angle.
Answer 2: Thank you. Unfortunately few studies with risk factors for POD were presented in literature. In the last years, some randomized trial was presented but the number of pts was small. For example, in the following study only 120 pts were randomized:
Subramaniam B, Shankar P, Shaefi S, Mueller A, O'Gara B, Banner-Goodspeed V, Gallagher J, Gasangwa D, Patxot M, Packiasabapathy S, Mathur P, Eikermann M, Talmor D, Marcantonio ER. Effect of Intravenous Acetaminophen vs Placebo Combined With Propofol or Dexmedetomidine on Postoperative Delirium Among Older Patients Following Cardiac Surgery: The DEXACET Randomized Clinical Trial. JAMA. 2019 Feb 19;321(7):686-696. doi: 10.1001/jama.2019.0234. Erratum in: JAMA. 2019 Jul 16;322(3):276. 
In addition, the following study from S. Korea is a retrospective analysis of 534 pts, but it was retrospective analysis 
Shin HJ, Choi SL, Na HS. Prevalence of postoperative delirium with different combinations of intraoperative general anesthetic agents in patients undergoing cardiac surgery: A retrospective propensity-score-matched study. Medicine (Baltimore). 2021 Aug 20;100(33):e26992.
On the other hand our analysis was prospective and we try to identify the risk factor for POD independ from anesthetic management only, while other study compared the effect of different anesthetic drug management to appearance POD.

Comment 3: There are several suggestions, and a strong recomandation for a more precise and scientific way to present your work.
Answer 3: Thank you for your comment. Our study is prospective study and the pts were enrolled in study before operation. Our results recorded in our database and data analysis were performed by SPSS software. In this study, we presented all perioperative details and characteristics of pts in tables. The risk factors for postoperative delirium were evaluated by two methods: compare two groups (with and without post-op delirium) of pts with parametric and non-parametric test and logistic regression analysis for identification the risk factors for acute post-op delirium. We think that our study meets all scientific criterias for data presentation and analysis.

Comment 4 Keywords should be 5, and contain the risk factors most important, because of your title.
Answer 4: Thank you. New key words were added in the manuscript
Change 4: New key words was added in second page

Comment 5: Very poor number of references. And choose, at least in the future works, literature since 2015. Only so, you can bring new into the new.
Answer 5: Thank you. Overall number of references is 29. More that 50% of references were form the last 12 years. In addition, few studies in past were reported with risk factors for post-op delirium.
From 2000-2010: 13
From 2010-2020: 10
From 2020-today: 6
The new of our study is that our study is prospective study and all perioperative available variables were included in analysis. On the other hand other studies (past and today) are focused to anesthetic drug management and not in overall course of pts postoperatively.

Comment 6: Study limitations must be placed before Conclusions.
Answer 6: Thank you. 
Change 6: The sections order was changed.

Comment 7: All the other declarations, I suggest they take a much scientific way. There is a unofficial way and a scientific way for writing of a paper work.
Answer 7: Thank you for your valuable comment. Our manuscript wrote with according guidelines of the Journal. In addition, our manuscript was divided for following sections: introduction, methods, results, conclusion, study limitations, references, tables.  All available methods today for data recording, presentation and analysis were implemented. Also, all section of manuscript was presented structurally. May be we have few weaknesses in our job, but one of the serious weaknesses is the small number of pts.

Comment 8: Material and methods: the inclusion criteria should be more complete and why not, include at least two risk factors cited in the introduction, such as electrolytes abnormal levels and familial predisposition for delirium and/or alcoholism.
Answer 8: Thank you. Inclusion criteria were presented in section methods and materials and they were: 
1) extubated patients,
 2) patients did not have known psychiatric disease, 
3) patients were speaking and understating the language 
4) age of patients >65 years old
About familial predisposition for alcoholism: it was not scope of our job. In addition, I think that familial predisposition for alcoholism does not exist. I am sorry! While predisposition for familial delirium it may be the factor which may affect the appearance delirium in general population. In our study we did not record and did not ask our pts if they had any person in their family with delirium history. Also, the pts which were included in our study had not any pre-op neurological and psychiatric dysfunctions. 
In section introduction, we mentioned in paragraph 4 the role of electrolytes disturbance for appearance the post-op delirium 

Comment 9: Important bias not mentioned in the exclsuion criterias and study limitations: degree of alcoholism.
Answer 9: Thank you. Our patients were not recording alcohol abuse, only social drinkers were included in the study. The pts who included in our study were not alcoholics.
Change 9: Changed was made in section ‘’study limitations’’.

Comment 10: Table 2 variables: you have to clasify the degree of the COPD, of the Chronic Renal Failure, of the smooking and drinking addiction.
Answer 10: Thank you. Chronic renal failure was defined as the 2-fold increase of plasma creatinine level compared with normal creatinine level. Smoking was defined current smoking and smoking any amount packs/year. COPD defined -long term use of bronchodilators and steroids for lung disease. Drinking addiction or abuse was not recorded in our pts, only alcohol use social drinkers were recorded. Other pts of our study did not use alcohol.

Comment 11: Please explain or  better reformulate the unhappy use for the variables: Profession( normal/sedentary) after MET.s; Education( primary, secondary, higher)- the meaning and the use is not very clear to me; Drug use- there is no use for mentioning it, if not the group of drugs are detailed and significant for the cardiac surgery necessity.
Answer 11: Thank you. Drug use-one patient received addictive substance. It did not affect any results. Education (primary, secondary, higher): higher=university, secondary (high school) =12-18 years old, primary=6-12 years old. Profession( normal/sedentary)=We think that normal work (which you proposed) is not suitable for division the type of work. We divided the type of work based the routine daily practice. Sedentary work was defined the work in any office, while the manual work was the work outside any office.  

Comment 12: Discussions: I think other variables should have been included, such as BMI, even because more than 50% are overweight.
Answer 12: Thank you. The BMI (even the 50% of pts was overweight) did not associate with post-op acute delirium (table 3) (p=0.57). Our discussion include many issue and I think that if the variables did not affect the appearance of POD, the mention and discuss it will be unnecessary information in section ‘’discussion’’. 

Comment 13: Conclusions: should be expressed more profesional and efficiently.
Answer 13: Thank you. We deleted the last sentence.
Change 13: Change was made 

Comment 14: did not understood or appreciate the formulation: " preoperative COPD"; "alcohol abuse": which is the limit, do you have some NASH Fibroscan evaluation of the alcoholic steatosis?.
Answer 14: Thank you. preoperative COPD=-long term use of bronchodilators and steroids for lung disease. Drinking addiction or abuse was not recorded in our pts, only alcohol use social drinkers were recorded. The pts who included in our study were not alcoholics, they were social drinkers. Other pts of  study did not use any amount alcohol.

Comment 15: Please explain the last phrase of the Conclusions, there is no sense and no meaning in it.
Answer 15: Thank you. We delete the last sentence.
Change 15: Change was made.

Reviewer 2 Report

As a general observation, the title and content of the paper does not fit very well in the Special Issue theme.

The study you have presented was mend to enlighten other risk factors for POD, than those already presented in the literature from years 2000-2010.

I expected from the team of authors to bring something new, or at least presented from another angle.

There are several suggestions, and a strong recomandation for a more precise and scientific way to present your work.

Keywords should be 5, and contain the risk factors most important, because of your title.

Very poor number of references. And choose, at least in the future works, literature since 2015. Only so, you can bring new into the new.

Study limitations must be placed before Conclusions.

All the other declarations, I suggest they take a much scientific way. There is a unofficial way and a scientific way for writing of a paper work.

Material and methods: the inclusion criteria should be more more complete and why not, include at least two risk factors cited in the introduction, such as electrolytes abnormal levels and familial predisposition for delirium and/or alcoholism.

Important bias not mentioned in the exclsuion criterias and study limitations: degree of alcoholism.

Table 2 variables: you have to clasify the degree of the COPD, of the Chronic Renal Failure, of the smooking and drinking addiction.

Please explain or  better reformulate the unhappy use for the variables: Profession( normal/sedentary) after MET.s; Education( primary, secondary, higher)- the meaning and the use is not very clear to me; Drug use- there is no use for mentioning it, if not the group of drugs are detailed and significant for the cardiac surgery necessity.

Discussions: I think other variables should have been included, such as BMI, even because more than 50% are overweight.

Conclusions: should be expressed more profesional and efficiently.

I did not understood or appreciate the formulation: " preoperative COPD"; "alcohol abuse": which is the limit, do you have some NASH Fibroscan evaluation of the alcoholic steatosis?.

Please explain the last phrase of the Conclusions, there is no sense and no meaning in it.

Author Response

(The authors gave the same response as above.)

Round 2

Reviewer 2 Report

Comment nr. 5 not adressed. Comment 10 not adressed.